# Short Communication: *age2exhume* - A Matlab/Python script to calculate steady-state vertical exhumation rates from thermochronometric ages and application to the Himalaya

Peter van der Beek[1*], Taylor F. Schildgen[2,1*]

[1] Institute for Geosciences, University of Potsdam, Potsdam, Germany.

[2] GFZ German Research Centre for Geosciences, Potsdam Germany.

* Both authors contributed equally to this work.

*Correspondence to*: Peter van der Beek (vanderbeek@uni-potsdam.de)

**Abstract.**

Interpreting cooling ages from multiple thermochronometric systems and/or from steep elevation transects with the help of a thermal model can provide unique insights into the spatial and temporal patterns of rock exhumation. Although several well-established thermal models allow for a detailed exploration of how cooling or exhumation rates evolved in a limited area or

along a transect, integrating large, regional datasets in such models remains challenging. Here, we present age2exhume, a thermal model in the form of a Matlab or Python script, which can be used to rapidly obtain a synoptic overview of exhumation rates from large, regional thermochronometric datasets. The model incorporates surface temperature based on a defined lapse rate and a local relief correction that is dependent on the thermochronometric system of interest. Other inputs include sample cooling age, uncertainty, and an initial (unperturbed) geothermal gradient. The model is simplified in that it assumes steady,

vertical rock-uplift and unchanging topography when calculating exhumation rates. For this reason, it does not replace more powerful and versatile thermal-kinematic models, but it has the advantage of simple implementation and rapidly calculated results. We also provide plots of predicted exhumation rates as a function of thermochronometric age and the local relief correction, which can be used to simply look up a first-order estimate of exhumation rate. In our example dataset, we show exhumation rates calculated from 1785 cooling ages from the Himalaya associated with five different thermochronometric

systems. Despite the synoptic nature of the results, they reflect known segmentation patterns and changing exhumation rates in areas that have undergone structural reorganization. Moreover, the rapid calculations enable an exploration of the sensitivity of the results to various input parameters, and an illustration of the importance of explicit modelling of thermal fields when calculating exhumation rates from thermochronometric data.

## 1. Introduction

The steady accumulation of thermochronometric data from around the world provides an opportunity to constrain spatial patterns of long-term (million-year timescale) exhumation with high granularity over vast swaths of the Earth's surface. This information can, in turn, provide clues to the driving mechanisms of orogen development and landscape evolution. Several well-established thermal models can be used to extract detailed cooling histories or exhumation rates from input cooling ages spread over a limited area or along an elevation transect. However, integrating information from large datasets, comprising cooling ages from multiple thermochronometers spread over a wide region, remains challenging due to the lack of easy-to-use tools that will handle such vast, multi-system datasets.

The most advanced modeling tools in common use by the thermochronology community include Pecube (Braun et al., 2012), HeFTy (Ketcham, 2005), QTQt (Gallagher, 2012), and GLIDE (Fox et al., 2014). Pecube is unique in its ability to handle forward and inverse thermal-kinematic modeling of spatially distributed data, including the options for time-varying topography as well as spatially and temporally variable rock-uplift driven by defined fault geometries and kinematics. This complexity, however, entails substantial set-up requirements and relatively high computational demands, which tend to limit the spatial extent of modeled datasets to $\sim 10^2$-$10^3$ km$^2$. HeFTy and QTQt, in contrast, model thermal histories only, for individual samples or samples that are assumed to fall into a pseudo-vertical alignment. GLIDE (Fox et al., 2014) was developed with the aim of extracting exhumation histories from regional datasets. While powerful, the temporally and spatially continuous coverage of calculated exhumation rates that the model produces requires interpolations that can be challenging to interpret without careful consideration of the spatial and temporal distribution of the input data (Fox et al., 2014; Schildgen et al., 2018).

Here we present a simple thermal model, age2exhume, which is optimized to provide a synoptic overview of exhumation rates from large regional datasets. This model, inspired by the original age2edot code (Brandon et al., 1998), takes the form of a Matlab or Python script that solves for steady-state exhumation rates from input thermochronometric ages, assuming vertical exhumation pathways and unchanging topography. A key difference between age2edot and age2exhume is that the former (despite its name) solves for ages given input exhumation rates, whereas our new model solves for exhumation rates given input ages. This difference makes age2exhume more suitable for calculating exhumation rates from regional datasets, since individual sample characteristics (e.g., an elevation-dependent surface temperature and local relief correction), included together with age in an input file, can be used to calculate an exhumation rate for each sample. A preliminary version of this code was used to visualize regional thermochronometric datasets in Schildgen et al. (2018); here, we provide more detailed background to the model and incorporate the individual sample characteristics mentioned above into the revised model.

The regional (constant) inputs to the model include crustal thermal properties that can be approximated or derived from the literature (an initial, unperturbed geothermal gradient, thermal model thickness, and thermal diffusivity) and kinetic parameters for the relevant thermochronometric systems, for which default values are provided. Sample-specific inputs include a local relief factor that can be extracted using standard GIS functions from a digital elevation model, elevation, thermochronometric

system, age, and age uncertainty. From our example dataset of 1785 cooling ages derived from five different thermochronometric systems in the Himalaya, steady-state, vertical exhumation rates with their uncertainties can be calculated within seconds on a standard laptop computer. Despite the synoptic nature of the results, we show how they reflect several fundamental features of the mountain belt, including strong regional differences that reflect known segmentation patterns and changing exhumation rates in areas that have undergone recent structural reorganization.

## 2. Background

### 2.1 Existing thermal models; their applications and limitations

Brandon et al. (1998) presented a simple, first-order approach to predict thermochronologic ages from input exhumation rates, in the form of a code called "age2edot". Age2edot calculates a steady-state conductive-advective geotherm and uses the approach of Dodson (1973) to predict the cooling-rate-dependent closure temperature of a given thermochronometric system. It then combines the predicted closure temperature and the steady-state geotherm to find the closure depth, and subsequently calculates a thermochronometric age by dividing the closure depth by the input exhumation rate. Kinetic parameters required for the Dodson (1973) calculation of closure temperature (see section 2.2 below) are derived from diffusion experiments for noble-gas based systems (i.e. (U-Th)/He and $^{40}Ar/^{39}Ar$) and from fitting an Arrhenius relation to experimental annealing data for fission-track systems (see Reiners and Brandon, 2006 for more detail). Simplifying assumptions in the age2edot approach include: (1) thermal steady state, (2) vertical exhumation paths, (3) unchanging topography, and (4) constant exhumation rates over the modelled time span. The most recent version of the age2edot code was released more than 15 years ago (Ehlers et al., 2005) and, because it was distributed as a Microsoft Windows executable, it is now obsolete.

Willett and Brandon (2013) published a modification to the age2edot approach, in which the steady-state geotherm solution was replaced by an (inherently transient) half-space solution, a correction for the sample elevation with respect to the regionally averaged elevation was introduced, and a best-fit exhumation rate is predicted from an input age and a modern (i.e., final) geothermal gradient. The code was provided as a Matlab script. Although it is computationally efficient, two aspects of this model limit its use for modelling large regional datasets in our view; one is of a practical nature, whereas the other is more fundamental. The practical limitation lies in the need to provide a value (or bounding values) for the modern geotherm for each prediction. Although this requirement makes conceptual sense, since only the modern geotherm can potentially be measured, it is of limited practical use because geothermal gradients are generally not known at more than very coarse spatial resolution, particularly in mountain belts. Moreover, the requirement is impractical when dealing with large datasets of widely varying ages, as geothermal gradients vary strongly in regions of variable exhumation rates. If the estimated bounding geotherms are poorly estimated (e.g., too low or high for a given thermochronometric age), no exhumation rate is returned. The more fundamental issue lies in the choice of a thermal half-space model, which leads to a strong sensitivity of the geotherm to exhumation rate and the persistence of transient thermal conditions even after several tens of millions of years of steady exhumation (Willett and Brandon, 2013). One type of data that allows assessing if, and how rapidly, thermal steady state might

be achieved in mountain belts is detrital thermochronology from sedimentary sequences in foreland basins. Several such datasets show constant lag times (i.e., thermochronometric age minus depositional age), interpreted as recording establishment of thermal steady state in the source area after only a few million years, including in the western European Alps (Bernet et al., 2001, 2009), the central and eastern Himalaya (Bernet et al., 2006; Chirouze et al., 2013), the eastern Himalayan syntaxis (Bracciali et al., 2016; Lang et al., 2016; Govin et al., 2020), Taiwan (Kirstein et al., 2010) and the Southern Alps of New Zealand (Lang et al., 2020). As argued by Bracciali et al. (2016), modelling these constant lag times using a thermal half-space model would require *decreasing* exhumation rates through time, with a rate of decrease that exactly offsets the transient upward advection of the geotherm, in all the above cases. More probably, these data indicate that the thermal half-space model is not ideal for representing orogenic geotherms.

A completely different approach Is taken by the thermal-history modelling codes HeFTy (Ketcham, 2005) and QTQt (Gallagher, 2012). These codes aim at predicting a thermal history from thermochronometric ages and additional measurements (in particular fission-track length distributions, but also kinetic indicators) for single samples, although the most recent versions of these codes allow modelling suites of vertically offset samples. The output of these models, when run in inverse mode, is an optimal time-temperature history and its uncertainty. These thermal history results require assumptions about the past geothermal gradient to be translated to a burial/exhumation history. Gallagher and Brown (1999) and Kohn et al. (2002) spatially interpolated thermal histories derived from large numbers of individual samples, using a precursor of the QTQt code, and combined them with heat-flow maps to derive regional to continental-scale images of denudation over geological time. This labor-intensive approach requires multiple thermochronometric systems and/or track-length data for each included sample in order to resolve meaningful thermal histories.

Pecube (Braun et al., 2012) is a three-dimensional thermal-kinematic code that predicts thermochronometric ages for various user-defined tectonic and geomorphic scenarios, taking into account the spatial and temporal perturbation of the geotherm by rock advection and transient topography. Pecube allows modelling both vertical and non-vertical exhumation paths, the latter controlled by a simple fault-kinematic model, and can be coupled to the neighborhood algorithm (Sambridge, 1999a, b) to run in inverse mode. The code has been used in a wide variety of tectonic and geomorphic settings (see Braun et al., 2012 for an overview), including at the scale of a small orogen (Curry et al., 2021). However, the fairly high computational demands of the code, particularly when run in inverse mode, make it best suited for models of more limited spatial extent (i.e., not exceeding several tens of km in length and width), where simple fault kinematics and/or spatially uniform rock-uplift can reasonably represent the tectonic deformation patterns.

GLIDE (Fox et al., 2014) comprises a linear inverse method to infer spatial and temporal variations in exhumation rate from spatially distributed thermochronometr datasets. GLIDE uses a numerical thermal model with a flux boundary condition at the base. The inversion assumes vertical exhumation and a smooth spatial variation in exhumation rates that can be described by a spatial correlation function. In this way, it uses exhumation constraints from one sample to help constrain exhumation in nearby regions, producing exhumation histories that are continuous in space and time. However, it has been argued that the code translates abrupt spatial variations in thermochronological ages, such as across faults, into temporal increases in

exhumation rates (Schildgen et al., 2018), unless the faults (or other features) are explicitly included in the correlation structure (Fox et al., 2014; Ballato et al., 2015). Willett et al. (2021) argued that such issues occur mainly in areas of insufficient data coverage without, however, quantifying this term; Schildgen et al. (2018) argued that most sampled regions on Earth with sharp spatial variations in exhumation have insufficient data coverage for unbiased prediction of exhumation-rate histories using GLIDE if those variations are not taken into account. From the above abbreviated review, we conclude that a simple, first-order method to assess large regional datasets in a consistent manner is currently lacking from the thermochronology toolbox. We aim to provide such a simple method with the age2exhume code.

## 2.2 Age2exhume method

Fig. 1 shows a sketch outline and flowchart for the age2exhume model. Input parameters for the model include the sea-level temperature $T_0$, atmospheric lapse rate $H$, the initial, unperturbed geothermal gradient $G_{init}$, thermal diffusivity $\kappa$, and model thickness $L$. The latter can represent the crustal thickness or, if appropriate, the maximum depth from which rocks have been exhumed, such as the depth to a regional detachment horizon. Input data for each sample include a thermochronometric age and its uncertainty at locations $x$ and $y$, sample elevation $h$, and local relief correction $\Delta h$. Kinetic parameters for the main low- to intermediate temperature thermochronometric systems (apatite and zircon (U-Th)/He and fission-track, mica $^{40}$Ar/$^{39}$Ar) are included as default values, but can be modified if desired.

When calculating exhumation rates from thermochronometric ages, a local relief correction ($\Delta h$) is needed to account for the difference in elevation of a sample ($h$) relative to an average-elevation ($h_{avg}$) surface that mimics the shape of the closure isotherm (Stüwe et al., 1994; Braun, 2002). We follow the procedure of Willett and Brandon (2013) in estimating the shape of that surface by averaging surface topography over a circle with a radius of $\pi \times z_c$, where $z_c$ is an estimated closure depth for the relevant thermochronometric system. A brief guide for how to implement this correction using a Digital Elevation Model in ESRI ArcMap or in QGIS is provided in Appendix A. The local relief correction $\Delta h$ is then calculated for each sample as:

$$\Delta h = h - h_{avg} \tag{1}$$

To predict a steady-state exhumation rate from a thermochronometric age, surface temperature, and the local relief correction, the model starts with an initial guess of the closure depth ($z_c$) and exhumation rate ($\dot{e}$) from an initial, unperturbed linear geothermal gradient ($G_{init}$), a nominal closure temperature ($T_c$), and a surface temperature ($T_s$):

$$z_c = \frac{(T_c - T_s)}{G_{init}} \tag{2}$$

$$\dot{e} = \frac{z_c + \Delta h}{age} \tag{3}$$

$T_s$ is calculated from an input sea-level temperature ($T_0$), the surface-temperature lapse rate ($H$), and the sample elevation at the position of $h_{avg}$: $T_{s(h)} = T_0 - H\, h_{avg}$. We use $h_{avg}$, rather than the actual sample elevation for this surface-temperature correction to simulate how surface temperature affects the thermal field at depth. For higher-temperature thermochronometers with deeper closure depths, $h_{avg}$ becomes more smoothed, and the associated impact of surface temperature on $z_c$ is reduced.

Note that the initial unperturbed geothermal gradient ($G_{init}$) is only used to calculate an appropriate basal temperature and to provide an initial estimate of the exhumation rate using equations (2) and (3).

The model then iteratively adapts $T_c$, $z_c$ and $\dot{e}$ until convergence to a steady-state solution. Importantly, $\Delta h$ is not recalculated after the initial estimate. Given the generally low sensitivity of $\Delta h$ to moderate variations in $z_c$, we believe this simplification is worthwhile, considering the consequent reduced computational demands. At each iterative step, first the advective

perturbation of the geotherm due to exhumation is calculated following Mancktelow and Grasemann (1997):

$$T_{(z)} = T_s + (T_L - T_s)\frac{(1 - e^{-z\dot{e}/\kappa})}{(1 - e^{-L\dot{e}/\kappa})} \tag{4}$$

where $T_{(z)}$ is the temperature at depth $z$, $T_L$ is the temperature at the base ($z = L$) of the model ($T_L = T_{avg} + G_{init} L$, where $T_{avg}$ is the temperature at the average elevation of the whole dataset), and $\kappa$ is the thermal diffusivity. Eq. (4) can be solved for the closure depth $z_c$:

$$z_c = z_{(T_c)} = \frac{\kappa}{\dot{e}} ln\left[1 - \frac{T_c - T_s}{T_L - T_s}\left(1 - e^{-L\dot{e}/\kappa}\right)\right] \tag{5}$$

Next, the closure temperature is re-estimated as a function of the cooling rate at the closure depth. First, the depth derivative of Eq. (4) is used to estimate the geothermal gradient:

$$\frac{dT}{dz} = \frac{\dot{e}(T_L - T_s)}{\kappa(1 - e^{-L\dot{e}/\kappa})} e^{-z\dot{e}/\kappa} \tag{6}$$

Eq. (6) is evaluated at the closure depth $z_c$. Because $\dot{e} = dz/dt$, the cooling rate ($\dot{T}$) is:

$$\dot{T} = \frac{dT}{dt} = \frac{dT}{dz}\dot{e} \tag{7}$$

The model then uses the Dodson (1973) equation to relate closure temperature to cooling rate:

$$T_c = \frac{E_a}{R \, ln\left(A\tau\frac{D_0}{a^2}\right)} \tag{8}$$

where $E_a$ (activation energy), $D_0$ (diffusivity at infinite temperature) and $a$ (diffusion domain size) are experimentally determined kinetic parameters for each thermochronological system, $A$ is a geometry factor and $\tau$ (characteristic time) is:

$$\tau = -\frac{RT_c^2}{E_a\dot{T}} \tag{9}$$

Once a new estimate for $T_c$ is obtained, $z_c$ is updated using Eq. (5) and a new estimate for the exhumation rate is obtained with Eq. (3). The model steps through equations (3) – (9) iteratively (Fig. 1b) until the change in exhumation rate between successive steps ($\Delta\dot{e}$) is smaller than a threshold value; here we use $|\Delta\dot{e}/\dot{e}| < 10^{-3}$. To ensure smooth convergence, the exhumation rate used in each successive step is the average between the previous and the newly calculated rate.

## 3. Results

### 3.1 General model predictions

Figs. 2 and 3 show contours of predicted exhumation rates for different combinations of age and $\Delta h$; Fig. 2 shows results for moderate exhumation rates ($< 2$ km Myr$^{-1}$) and thermochronometric ages up to 30 Ma, whereas Fig. 3 zooms in on the youngest ages ($< 5$ Ma) and shows results for exhumation rates up to 5 km Myr$^{-1}$. Input parameters for these models are as in Table 1, except that a constant surface temperature ($T_s$) of 10 °C was used, because absolute sample elevation is not included in these generic models. Kinetic parameters for the apatite (U-Th)/He (AHe) system are derived from Farley (2000); for the zircon (U-Th)/He (ZHe) system from Reiners et al. (2004); and for the apatite (AFT) and zircon (ZFT) fission-track systems from Reiners and Brandon (2006). These results can be thought of conceptually as showing age – elevation profiles for different constant exhumation rates, with elevation measured relative to an average regional elevation as defined in Section 2.2. They can also be used as a plotted lookup table for rapidly inferring exhumation rate from a given age, $\Delta h$ combination.

### 3.2 Results from a Himalayan example data set

Our example data set from the Himalaya comprises 1785 thermochronologic ages compiled from papers published through July 2022; data sources are provided in the Supplementary Information. We have excluded some reported ages from the Siwaliks (Sub-Himalayan fold-thrust belt), as that sedimentary unit commonly yields unreset ages. We have also excluded the western and eastern syntaxis regions, where extremely rapid exhumation is driven by processes that are different from those in the main part of the Himalaya (Zeitler et al., 2014; Butler, 2019). Finally, we exclude any pre-Himalayan ages ($> 60$ Ma), as these are not directly linked to exhumation during Himalayan mountain building. Our data set comprises 345 white mica $^{40}$Ar/$^{39}$Ar (MAr) ages, 236 ZFT ages, 783 AFT ages, 281 ZHe ages, and 140 AHe ages. All ages and sample details are included in a single Excel file, with columns that include a sample ID number, latitude, longitude, elevation, $\Delta h$ value, age, 1σ age uncertainty, and a numeric code for the thermochronologic system (Schildgen and van der Beek, 2022a). Table 1 shows the parameters we assume for the surface temperature ($T_0$, $H$) and the thermal model ($L$, $G_{init}$, $\kappa$). Kinetic parameters used for the AHe, AFT, ZHe and ZFT systems are the same as for the general model predictions presented in section 2.1 above; we used the parameters from Hames and Bowring (1994) for the MAr system.

A map of the calculated exhumation rates for the Himalaya (Fig. 4) shows exhumation plotted such that rates derived from lower-temperature systems plot on top of those from higher-temperature systems. When the symbol for a lower-temperature system is darker than the symbol of a higher-temperature system plotted below it, this implies that exhumation rates have slowed through time. Conversely, a lighter color for the lower-temperature system plotted over a higher-temperature system implies exhumation rates have increased through time. The map reveals patterns in space and time that reflect well-known structural patterns of the range. In general, a band of rapid exhumation rates occurs at the topographic front of the high Himalaya, with slower rates recorded to the north and south. Within this band, the highest rates are generally recorded by the lower-temperature AHe and AFT thermochronometers, suggesting increasing exhumation rates with time. Note that such

variable exhumation rates recorded by different co-located thermochronometers formally violate the assumption of constant exhumation rates through time implicit in the model. The rates inferred from the higher-temperature thermochronometers should therefore be considered rough estimates only; they will generally be overestimated in the case of increasing rates through time, and the corresponding rate change will therefore be underestimated. The focused rapid rates at the foot of the high Himalaya together with an increase in exhumation rates for lower-temperature systems is consistent with exhumation being driven by thrusting over a large-scale ramp in the Main Himalayan Thrust (MHT), the interface between the underthrusting Indian continent and the overlying Himalayan units, often associated with duplex development (e.g., Robert et al., 2009; Herman et al., 2010; Coutand et al., 2014; Dal Zilio et al., 2021; van der Beek et al., in press).

The highest exhumation rates (> 2 km Myr$^{-1}$) outside of the Himalayan syntaxes occur in central Nepal (~84 ºE), Sikkim (~88 ºE), the Kumaun Himalaya (~80 ºE), and the Sutlej valley (~78 ºE). High rates (between 1 and 2 km Myr$^{-1}$) are recorded along the high Himalayan front throughout northwest India (~76-80 ºE) and more sporadically in eastern Nepal (~87 ºE) and western Bhutan (~89 ºE). The lowest exhumation rates along the high Himalayan topographic front (< 0.8 km Myr$^{-1}$) are found in Kashmir (west of ~75 ºE), western Nepal (~81 ºE), and from western Bhutan (~90 ºE) to the east. These lateral variations in exhumation rates have been interpreted as reflecting lateral variations in the presence/absence and geometry (location, height and dip) of the mid-crustal ramp in the MHT, together with duplex formation and local out-of-sequence thrusting (Hubbard et al., 2021; Dal Zilio et al., 2021; van der Beek et al., in press). In some of the more slowly exhuming regions, in particular in Bhutan, exhumation rates appear to be decreasing through time, with lower-temperature systems recording lower exhumation rates than higher-temperature systems. Decreasing exhumation rates in Bhutan can be linked to slowing convergence across the Bhutan Himalaya due to transfer of deformation to the Shillong Plateau to the south (Clark and Bilham, 2008; Coutand et al., 2014, 2016). Similar to the caveats described above concerning increasing exhumation rates, in areas of decreasing exhumation rates, the change in rates through time recorded by different systems will also be underestimated.

The above example illustrates how this method can rapidly provide internally consistent estimates of exhumation rates from multiple thermochronometers from different elevations over a large region. Inferred patterns of exhumation rates can be linked to structural and geophysical observations of orogen segmentation, as above, or to orogen-wide topographic measures for assessing first-order linkages between exhumation rates and morphology (e.g., Clubb et al., 2022).

## 4. Discussion and Conclusions

### 4.1 Importance, uncertainties and sensitivity

An advantage of the rapid calculations performed by age2exhume is that it is easy to explore the sensitivity of the calculated exhumation rates to different input parameters (i.e., sample-specific information and crustal/thermal properties), in addition to evaluating how the iterative method compares to simpler estimates of exhumation rates. Regarding the latter, we can compare calculated exhumation rates from age2exhume to those that would be obtained by assuming a simple linear geotherm and fixed nominal closure temperature, $T_c$. Fig. 5a compares "initial" exhumation rates, calculated using Eqs. 2 and 3 (hence, a linear

geotherm and fixed $T_c$), with the final exhumation rates predicted by age2exhume, which incorporate perturbations to the geotherm and $T_c$. Initial exhumation rates are calculated using the same thermal parameters of Table 1 and nominal closure temperatures of 70 °C for the AHe system, 120 °C for the AFT system, 180 °C for the ZHe system, 220 °C for the ZFT system, and 350 °C for the MAr system. The comparison shows that for exhumation rates up to ~0.5 km Myr$^{-1}$, there is little difference between the two methods (Fig. 5a). At higher exhumation rates, the methods deviate substantially, with the initial estimate systematically overestimating the exhumation rate. For example, at exhumation rates ≥2 km Myr$^{-1}$, overestimates mostly fall between 100 and 300%. These findings can be explained by considering the relative importance of two competing influences on the closure depth $z_c$ (Fig. 1a), which directly determines the exhumation rate (Eq. 3). On one hand, higher cooling rates – linked to higher exhumation rates – lead to an increase in $T_c$, and hence a deepening of $z_c$ (Eqs. 8, 9). On the other hand, the advective perturbation of the geotherm due to exhumation, which forces an upward deflection of isotherms, leads to a shallowing of $z_c$ for any $T_c$ (Eq. 5). The degree of advective perturbation of the geotherm is characterized by the non-dimensional Péclet number: $Pe = \dot{e}\,L/\kappa$ (e.g., Braun et al., 2006); the predicted (surface) geothermal gradient thus increases with increasing exhumation rate (Fig. 6). With higher exhumation rates, the effect of upward, advective perturbation of isotherms on $z_c$ dominates over the effect of the increasing $T_c$ on $z_c$. The scatter in the amount of overestimation, in particular for the lower-temperature AFT and AHe systems, is linked to the effect of including $\Delta h$, which is more important for shallower $z_c$ (Eq. 3).

But how important are these differences in the method of calculating exhumation rates relative to the uncertainties in any calculated rate? The uncertainties in reported ages are just one component of the total uncertainty that one can consider in an exhumation-rate calculation, but the direct propagation of age uncertainty into the uncertainty on an inferred exhumation rate provides a simple means of comparison (Fig. 5b). Because of the non-linear relationship between age and exhumation rate, the uncertainties in exhumation rates are asymmetric, with $\dot{e}_{max} - \dot{e} > \dot{e} - \dot{e}_{min}$. The bulk of the relative uncertainties in exhumation rates associated with age uncertainty lie between 10 and 50%, and they are not strongly dependent on exhumation rate. Higher-temperature systems (ZHe, ZFT and MAr) are generally associated with lower exhumation-rate uncertainties (< 10%) because of the smaller age uncertainties associated with these systems. In contrast, AFT data can have uncertainties of up to >100%, because low track counts due to low U-contents and/or young ages yield large age uncertainties. Some large relative uncertainties in the AHe and ZHe systems at lower exhumation rates (< 1 km Myr$^{-1}$) are probably associated with larger inter-grain scatter in ages due to compositional and grainsize effects that become more important at lower cooling and exhumation rates (e.g., Whipp et al., 2022 and references therein). Overall, however, the bulk of the uncertainties in exhumation rate are smaller than the differences between the initial and final exhumation rates shown in Fig. 5a for exhumation rates > ~0.5 km Myr$^{-1}$. This comparison implies that the thermal effects of exhumation significantly affect inferred exhumation rates in tectonically active areas.

The importance of including sample-specific information in exhumation-rate calculations is illustrated in Figs. 7a and 7b. Our comparison of exhumation rates calculated with a constant surface temperature, $T_s$, versus those calculated with $T_s$ dependent on elevation shows a relatively small effect, with differences mostly less than 10%. However, for the low-temperature

thermochronometers AHe and AFT at exhumation rates < 1 km Myr$^{-1}$, differences can reach 20% (Fig. 7a). The effect of the local relief correction, $\Delta h$, for each sample is generally more important. Although the magnitude of $\Delta h$ tends to be reduced for the lower-temperature systems (because their closure isotherms more closely mimic surface topography), any given $\Delta h$ has a stronger impact on exhumation rates for low-temperature systems (with shallower $z_c$) compared to high-temperature systems (Fig. 7b; Eq. 3). Moreover, the effects are asymmetric: negative $\Delta h$ values lead to a larger correction in exhumation rates compared to positive $\Delta h$ values. For example, a $\Delta h$ of +1 km will lead to a ca. 20% change in calculated exhumation rate for the AFT system, whereas a $\Delta h$ of -1 km will lead to a 30 to 50% change (Fig. 7b). This asymmetry results from the non-linear effect of exhumation rate on $z_c$: positive $\Delta h$ values will lead to increased predicted exhumation rates, but these increases will be partly offset by the resulting advective perturbation of the geotherm. In contrast, the decreased predicted exhumation rates for negative $\Delta h$ values will be less modified by advective effects. The importance of including $\Delta h$ when calculating exhumation rates is further emphasized when considering that samples are more commonly collected from valley bottoms (with negative $\Delta h$ values) than ridgetops. Our Himalayan example dataset bears this out: the histogram of $\Delta h$ values is skewed toward negative values, with a median $\Delta h$ of -0.53 km (Fig. 7b inset).

We next explore the sensitivity of calculated exhumation rates to crustal parameters, including the model thickness $L$ (Fig. 7c) and the initial, unperturbed geothermal gradient $G_{init}$ (Fig. 7d) . These plots show the percent change in predicted exhumation rates when changing these two parameters to either a higher or a lower value relative to the default values of $L$ = 30 km and $G_{init}$ = 25 °C/km. Decreasing $L$ from 30 to 20 km leads to higher predicted exhumation rates (by up to $\sim$ 40 %), whereas increasing $L$ from 30 to 40 km leads to lower predicted exhumation rates (by up to $\sim$ -20 %), with the magnitude of the effect increasing with exhumation rate (Fig. 7c). This behavior can be understood by considering the effect $L$ has on the advective perturbation of the geotherm, through the Péclet number (see above): the Péclet number is linearly dependent on $L$ so that, for constant exhumation rate $\dot{e}$ and diffusivity $\kappa$, increasing $L$ will lead to a stronger perturbation of the geotherm and thus a shallower closure depth for any thermochronometer. The sensitivity of the predictions to $G_{init}$ is of similar magnitude when considering changes from 25 to 30 °C/km or from 25 to 20 °C/km (Fig. 7d), but in this case, the effect is strongest for relatively low exhumation rates and thus relatively unperturbed geothermal gradients.

## 4.2  Limitations and recommended use

The assumptions underlying the model limit its use, strictly speaking, to settings where both topography and exhumation rates are temporally constant throughout the time-period considered and exhumation is mainly vertical. The requirement for constant exhumation rates is due to both the use of a steady-state solution for the advectively perturbed geothermal gradient and the use of the Dodson (1973) approach to estimate closure temperatures. These assumptions break down in cases where thermal and exhumation histories are more complex, in particular when they include phases of burial and heating. However, as our Himalayan example shows, our approach can provide first-order information on spatio-temporal patterns of exhumation, highlighting regions of accelerating versus decelerating exhumation. This result comes with the caveat that the model will systematically underestimate the change in exhumation rate, as discussed above. The Himalayan example also shows that first-

order results can be obtained in a setting where horizontal advection of rocks is important; however, in this case, the interpretation of possible accelerations or decelerations in exhumation rate should take the regional structure and kinematics into account. For instance, accelerated exhumation may be due to rocks moving over a flat-to-ramp transition in a crustal-scale decollement, rather than a temporal change in tectonic or climatic drivers.

Our analysis of the importance of including advective perturbation of the geotherm in the thermochronometric age predictions shows that this effect is not significant for exhumation rates $<\sim 0.5$ km Myr$^{-1}$ (Fig. 5a). At these relatively low rates of exhumation and cooling, kinetic effects also become important in controlling thermochronometric ages (e.g., Whipp et al., 2022) and these are not included in our model. The model is thus best suited for the analysis of regional datasets from rapidly and continuously exhuming regions, i.e., tectonically active mountain belts. The relief correction included in the model makes

it suitable to handle data that were collected at widely varying elevations.

The model assumptions of a constant basal temperature together with an input model thickness are unlikely to be valid over long timescales, and in many cases can only be estimated roughly. However, the speed with which exhumation rates can be calculated from our model enables users to easily investigate the sensitivity of their results to these estimated values. Moreover, while these thermal parameters change the absolute values of the predicted exhumation rates, they affect all predictions

similarly (if not equally). Therefore, their influence on spatial patterns in exhumation rates or the correlation of exhumation rates with other metrics will be limited.

We provide three different versions of the  model in the form of Matlab scripts on Zenodo (van der Beek and Schildgen, 2022): (1) a basic version that takes a single age – $\Delta h$ pair as input and returns a single exhumation rate; (2) a version for which a range of thermochronologic ages and $\Delta h$ values are provided and that returns a lookup table of exhumation rates (used in

Section 3.1 and Figs. 2, 3); and (3) a version that reads an input file of sample locations, elevation, thermochronologic system, age and uncertainty, and returns a table of exhumation rates with uncertainty, closure depths, and surface steady-state geotherms for each sample (used in section 3.2 and Fig. 4). A correctly formatted input file is also included in the Zenodo repository. We anticipate the latter version to be most useful, and therefore we also provide a Python script with that same functionality (Schildgen and van der Beek, 2022b). Alternatively, Figures 2 and 3 of this paper can be used to simply look up

appropriate exhumation rates for a given age – $\Delta h$ combination, but note that these figures are plotted for particular values of the input parameters $G_{init}$, $L$ and $\kappa$.

### 4.3  Concluding remarks

The model presented here, age2exhume, enables a first-order, synoptic view of spatial and temporal variations in exhumation rates, calculated in a rapid, self-consistent manner from different thermochronometers. The main advantage of our approach

over the version of age2edot presented by Willett and Brandon (2013) is that our model does not require the final geothermal gradient as input, but only the initial, unperturbed geotherm. This aspect of our model makes it easily applicable to regions with strongly varying exhumation rates, which are expected to have a wide range of modern geothermal gradients. The modern

geothermal gradient, when known, adds an additional constraint to the model solution. However, for many regions of the world, particularly for mountain belts, modern geothermal gradients are essentially unknown. In our entire Himalayan study region, for instance, the global heat-flow database (https://ihfc-iugg.org; interrogated 10/10/2022) does not contain a single data point. Although there are data both for the Tibetan Plateau to the north and the Ganges foreland basin to the south, these are not useful for assessing the perturbed geotherm within the mountain belt. Our model does provide the predicted steady-state surface geotherm as output, so it can be compared to any potential measurements (Fig 6).

Our model assumes steady-state exhumation, unchanging topography, and vertical exhumation pathways, so it is only appropriate for obtaining first-order, synoptic overviews of exhumation-rate patterns in regions of relatively rapid, continuous exhumation. Nevertheless, in the case where ages from multiple thermochronometers are available from individual samples or from samples in close proximity to one another, differences in exhumation rates derived from those ages can be used to map out where changes in exhumation rates have likely occurred, and thus highlight regions where more advanced thermal modeling could be used to extract non-steady-state exhumation histories. The rapidity with which our model calculates regional patterns of exhumation rates also allows testing its sensitivity to the different input parameters.

## Appendix A: Calculating Δh from digital elevation datasets

To calculate $\Delta h$, Willett and Brandon (2013) suggest calculating a mean value for a circle that has a radius equal to $\pi \times z_c$, where $z_c$ is the closure depth of the system. This calculation can be done with standard operations in a geographic information system (GIS), or other tools designed to work with continuous raster datasets. The following instructions can be followed to calculate $\Delta h$ values within ArcMap from ESRI (version 10.8.1) or within QGIS (version 3.26). We have not tested if the instructions are easily applicable to earlier versions of the software. Nevertheless, small modifications to these procedures can likely be found by searching on the names of the functions described below. Importantly, regardless of the software package used to calculate $\Delta h$, the spatial extent of the DEM should extend beyond the limits of the sample points, with a buffer zone at least equal to the highest radius that will be considered. For example, the DEM should extend at least ca. 10 km beyond the spatial extent of the sample data to prevent edge effects from affecting $\Delta h$ calculations for the MAr system.

*ESRI ArcMap*

In ESRI's ArcMap version 10.8.1, the mean elevation can be calculated using the Focal Statistic function, found within the "Spatial Analyst Tools - Neighborhood" tools in Arc Toolbox. The Focal Statistic function provides an option to average values over a moving circular window with a radius defined by map units or by a number of pixels. For example, for a standard 90-m resolution SRTM DEM, and for a desired $z_c$ of 2000 m (e.g., for the AHe system), the radius of the circle should be 6280 m, which is approximately equivalent to 70 pixels. To efficiently calculate $\Delta h$ for all samples in a large dataset, it is practical to take advantage of the "Raster Calculator" (Spatial Analyst Tools – Map Algebra) and the "Extract Values to Points" functions (Spatial Analyst Tools – Extraction). The Map Calculator can be used to subtract the smoothed DEM from the previous step from the modern DEM. This operation will produce a continuous raster data set of $\Delta h$ values. The "Extract

Values to Points" function samples a raster at the position of each sample data point, and adds the extracted value to a new column ("field") in the attribute table of the shapefile. Although the exact procedure described here may differ for other versions of ArcMap, general functions to calculate focal statistics, perform arithmetic operations on raster datasets, and automated extraction of values from rasters at the location of sample points can be found in many versions of the software.

*QGIS*

In QGIS 3.26, a procedure to find the average elevation over a defined circular search area can be accomplished with the SAGA plug-in, which can be installed directly from the "Plug-in" menu and then "Manage and install plug-ins". After installation, the SAGA tools can be found within the "Processing" menu, then "Toolbox". Within SAGA, go to the "Raster Filter" options and then select "Simple filter". The Filter option should be set to "Smooth" (to calculate an average value), and the Kernel type set to "Circle". The radius should be set to the number of pixels that will provide the correct radius length.

Like in the example above, for a standard 90-m resolution DEM and a desired $z_c$ of 2000 m (for the AHe system), the radius of the circle ($\pi \times z_c$) should be 6280 m, which is approximately equal to 70 pixels. Next, the Raster Calculator within QGIS can be used to calculate $\Delta h$ values over the extent of the DEM, by subtracting the smoothed DEM calculated in the previous step from the original DEM. Finally, to extract the calculated $\Delta h$ values for each sample point, within the standard Processing Toolbox under the "Raster analysis" heading is the function "Sample raster values". With this tool, the point layer containing

the sample points should be given as the "Input layer" and the raster of $\Delta h$ values should be given as the "Raster layer". The output point file includes all of the attributes of the original point layer, but adds a column containing the extracted $\Delta h$ value for each point. That file, by default, is only saved to memory. To save it permanently, the small square-shaped icon to the right of the layer name can be clicked to bring up a dialog box that allows saving the file to a defined location.

**Code Availability.** The Matlab scripts for three versions of the age2exhume code, together with an input file, are included in the Zenodo repository: age2exhume Matlab scripts (https://doi.org/10.5281/zenodo.7341603). The Python version of age2exhume, together with an input file, can be downloaded from the Zenodo repository: age2exhume python script (https://doi.org/10.5281/zenodo.7341690).

**Data Availability.** Data used in the example data set was compiled from the sources listed in the Supplementary Material. An

405 Excel file containing the full dataset and calculated exhumation rates is included in the Zenodo repository: Thermochronology dataset for Himalaya (https://doi.org/10.5281/zenodo.7053115).

**Author Contributions.** Both authors contributed equally to the development of the code, the compilation of the Himalayan dataset, sensitivity analyses, and writing of the manuscript. T.S. wrote the Python version of the code.

**Competing Interests.** The authors declare no competing interests.

**Acknowledgements.** T.S. acknowledges support from the ERC Consolidator Grant #863490 GyroSCoPe; P.v.d.B. acknowledges support from the ERC Advanced Grant #834271 COOLER. We thank reviewers Matthew Fox and David Whipp for constructive comments that improved the quality of the manuscript and allowed us to refine some aspects of the code, and Associate Editor Melissa Tremblay for efficient handling of the manuscript. We also thank David Whipp for his freely accessible online "Geo-Python" course, which provided the base training needed for writing the Python version of the code.

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

**Figure captions**

Figure 1: Model outline. (a) Sketch of model showing some of the main model parameters; main plot is a temperature – depth $(T-z)$ plot of the model domain, showing initial, unperturbed linear geotherm ($G_{init}$) and initial estimates of closure temperature ($T_c$) and closure depth ($z_c$) in grey, and final, steady-state advectively perturbed geotherm and calculated $T_c$ and $z_c$ in black. Note that in most cases, $T_c$ will increase because of the increased cooling rate (Eqs. 8, 9), while $z_c$ will decrease due to the advective perturbation of the geotherm (Eq. 5). Inset shows how the local relief correction $\Delta h$ is derived from the relationship between sample elevation (indicated by black dot) and average elevation $h_{avg}$. (b) Flow chart of the model and its main iteration loop. Abbreviations for input parameters are explained in the main text.

Figure 2: Contour plots of exhumation rate for different age – $\Delta h$ combinations. These can be thought of as age – elevation relationships for different constant exhumation rates. Plots are shown for the (a) AHe, (b) AFT, (c) ZHe, and (d) ZFT systems; exhumation-rate contours are shown every 0.05 km Myr$^{-1}$ from 0 to 2.0 km Myr$^{-1}$.

Figure 3: Contour plots of exhumation rate for different age – $\Delta h$ combinations, zooming in on rapid rates and young thermochronologic ages (< 5 Ma). Plots are shown for the (a) AHe, (b) AFT, (c) ZHe, and (d) ZFT systems; exhumation-rate contours are shown every 0.1 km Myr$^{-1}$ from 0 to 5 km Myr$^{-1}$.

Figure 4: Exhumation rates inferred from Himalayan dataset of 1785 thermochronologic ages. Each data point represents a time-averaged exhumation rate associated with a thermochronometric age. (a) Western Himalaya (Kashmir to Nepal); (b) Nepal Himalaya; (c) Eastern Himalaya (Sikkim to Arunachal Pradesh). Data points are coloured by exhumation rate; symbols indicate different thermochronometric systems (see legend in (a)). Inset in (a) shows the locations of the three maps within the Himalaya.

Figure 5: Impact of including perturbations to the geotherm and $T_c$ in estimates of exhumation rate, and uncertainties in exhumation-rate calculations. (a) Comparison of initial exhumation rate ($\dot{e}_{init}$; assuming a linear geothermal gradient and nominal closure temperatures) for the Himalayan data against final exhumation rate ($\dot{e}$), calculated using the age2exhume method. The impact is expressed as a percent change between the two results; i.e., $100 \times (\dot{e}_{init} - \dot{e})/\dot{e}$. Symbols indicate different thermochronometric systems. (b) Relative uncertainty in exhumation rate calculated by propagating uncertainty in age. Symbols are as in (a). Inset shows stacked histograms of relative uncertainty for different systems. See text for discussion.

Figure 6: Predicted surface geothermal gradient as a function of predicted steady-state exhumation rate for all Himalayan data. Input linear geotherm $G_{init} = 25$ °C/km.

Figure 7: Impact of varying surface conditions and sensitivity to thermal parameters on calculated exhumation rates. (a) Impact of using a variable (elevation-dependent) surface temperature versus a constant surface temperature; (b) impact of including the local relief correction $\Delta h$; inset shows histogram of $\Delta h$ values for the Himalayan dataset. Plots in a and b show percent change in exhumation rates when the corrections are not included compared to when they are included, i.e., $100 \times (e_{const.\ Ts} - e_{variable\ Ts})/e_{variable\ Ts}$ and $100 \times (e_{without\ \Delta h} - e_{with\ \Delta h})/e_{with\ \Delta h}$. (c) Sensitivity of predicted exhumation rates to model thickness $L$; (d) sensitivity of predicted exhumation rates to initial, unperturbed geothermal gradient $G_{init}$. Plots in (c) and (d) show percent change in exhumation rates for varying conditions versus exhumation rate predicted with parameters of Table 1; i.e., $100 \times$ (tested change – default value)/default value, where "default value" is defined as in Table 1. See text for discussion.

| Parameter | Symbol | Value | Unit |
|---|---|---|---|
| Temperature at sea level | $T_0$ | 25 | °C |
| Atmospheric lapse rate | $H$ | 5 | °C km$^{-1}$ |
| Initial geothermal gradient | $G_{init}$ | 25 | °C km$^{-1}$ |
| Thermal diffusivity | $\kappa$ | 30 | km$^2$ Myr$^{-1}$ |
| Model thickness | $L$ | 30 | km |

Table 1: input parameter values used in modelling Himalayan dataset

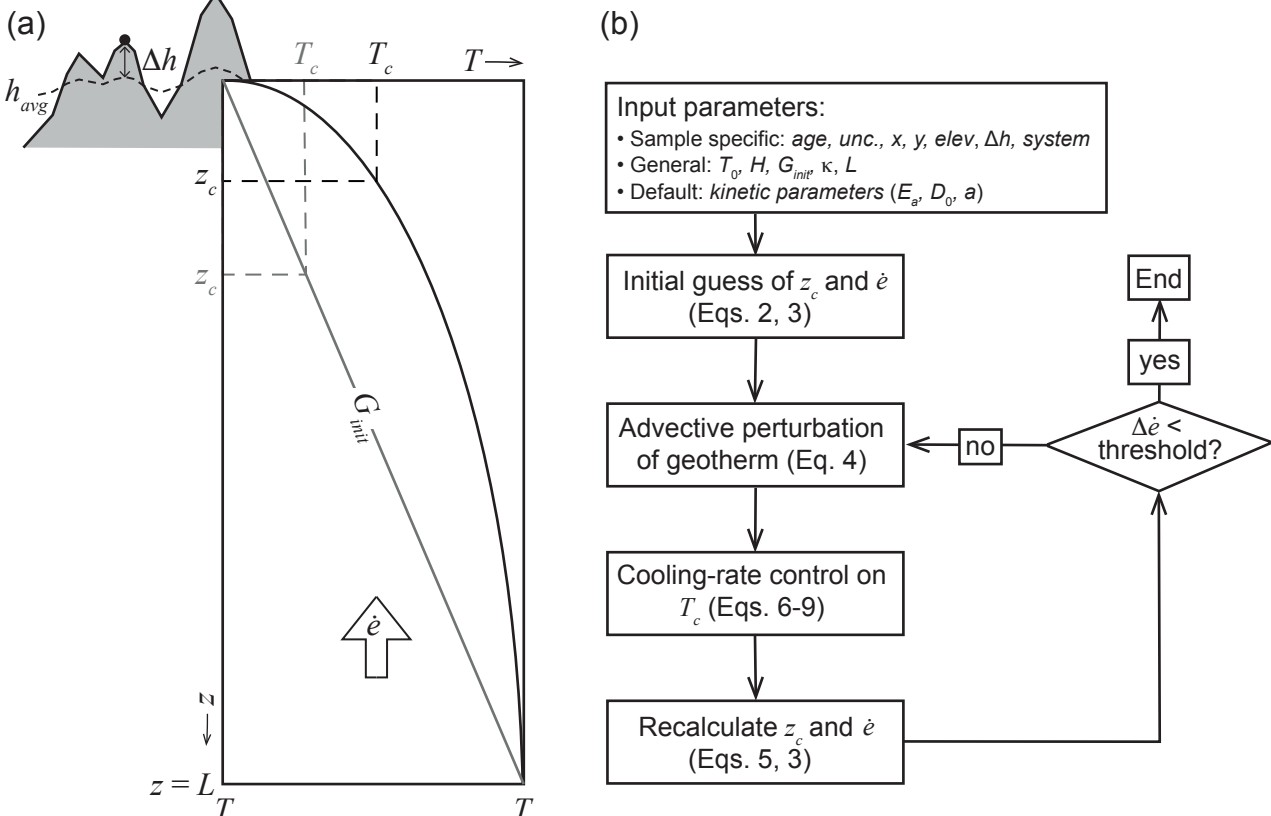

van der Beek and Schildgen
Figure 1

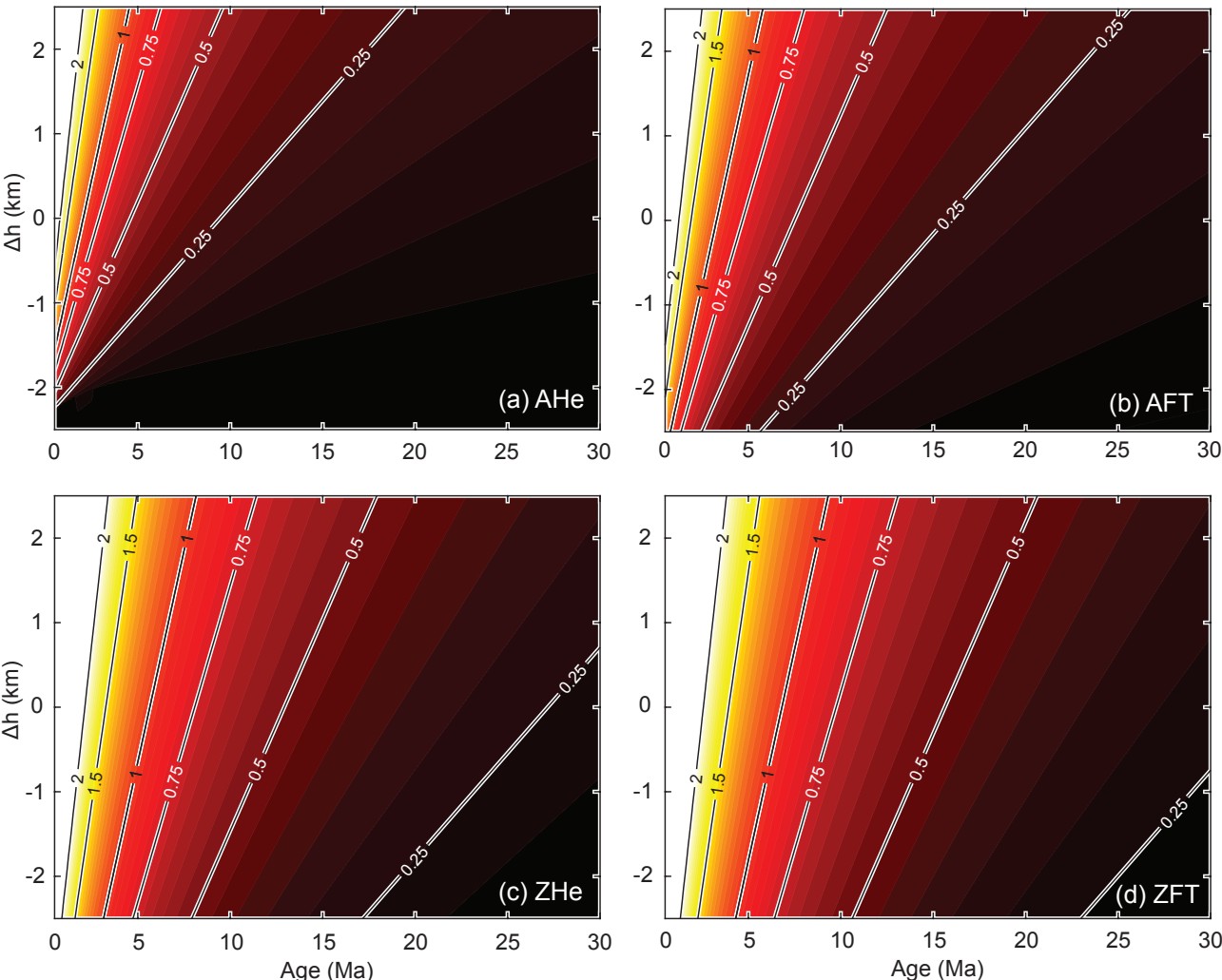

van der Beek and Schildgen
Figure 2

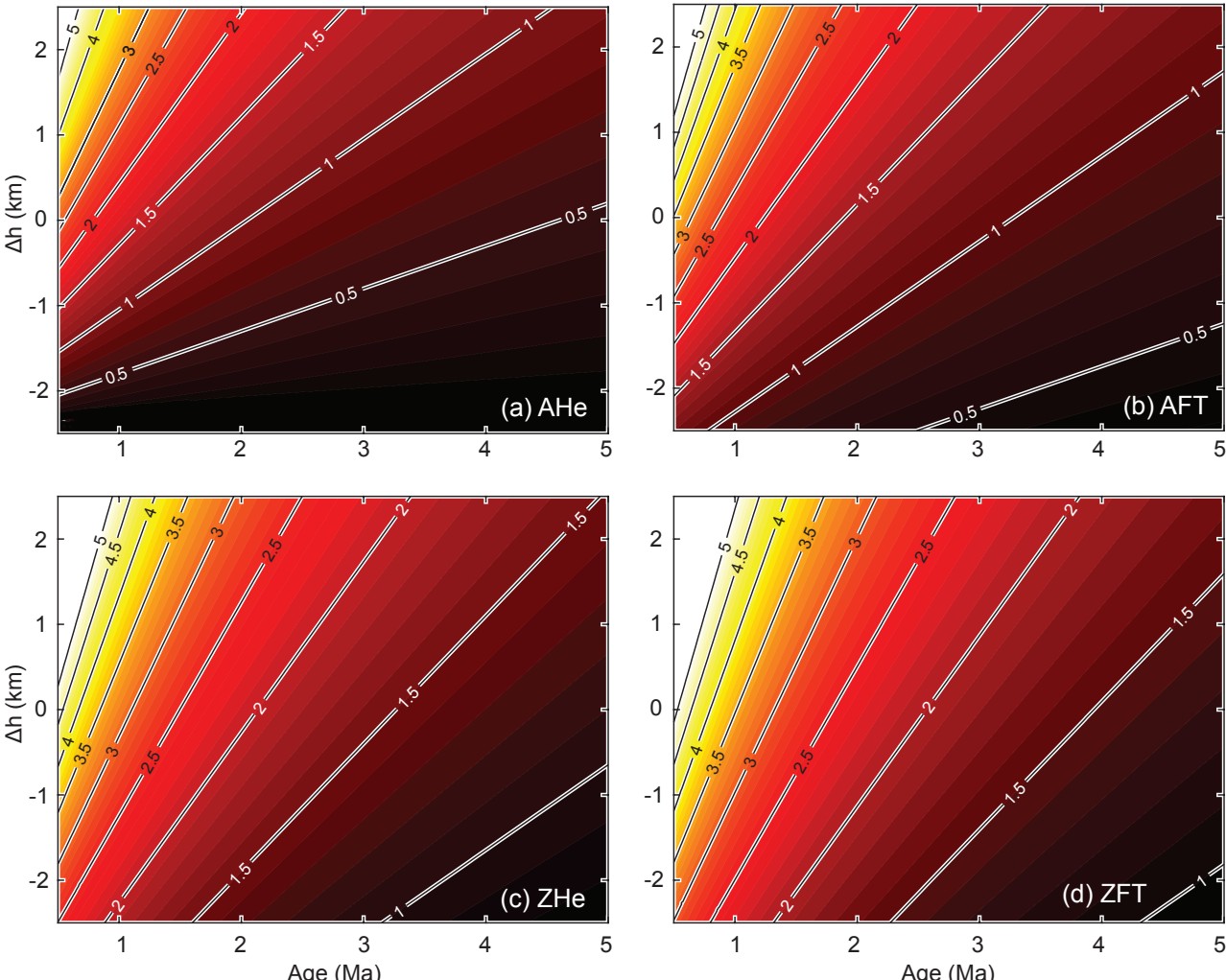

van der Beek and Schildgen
Figure 3

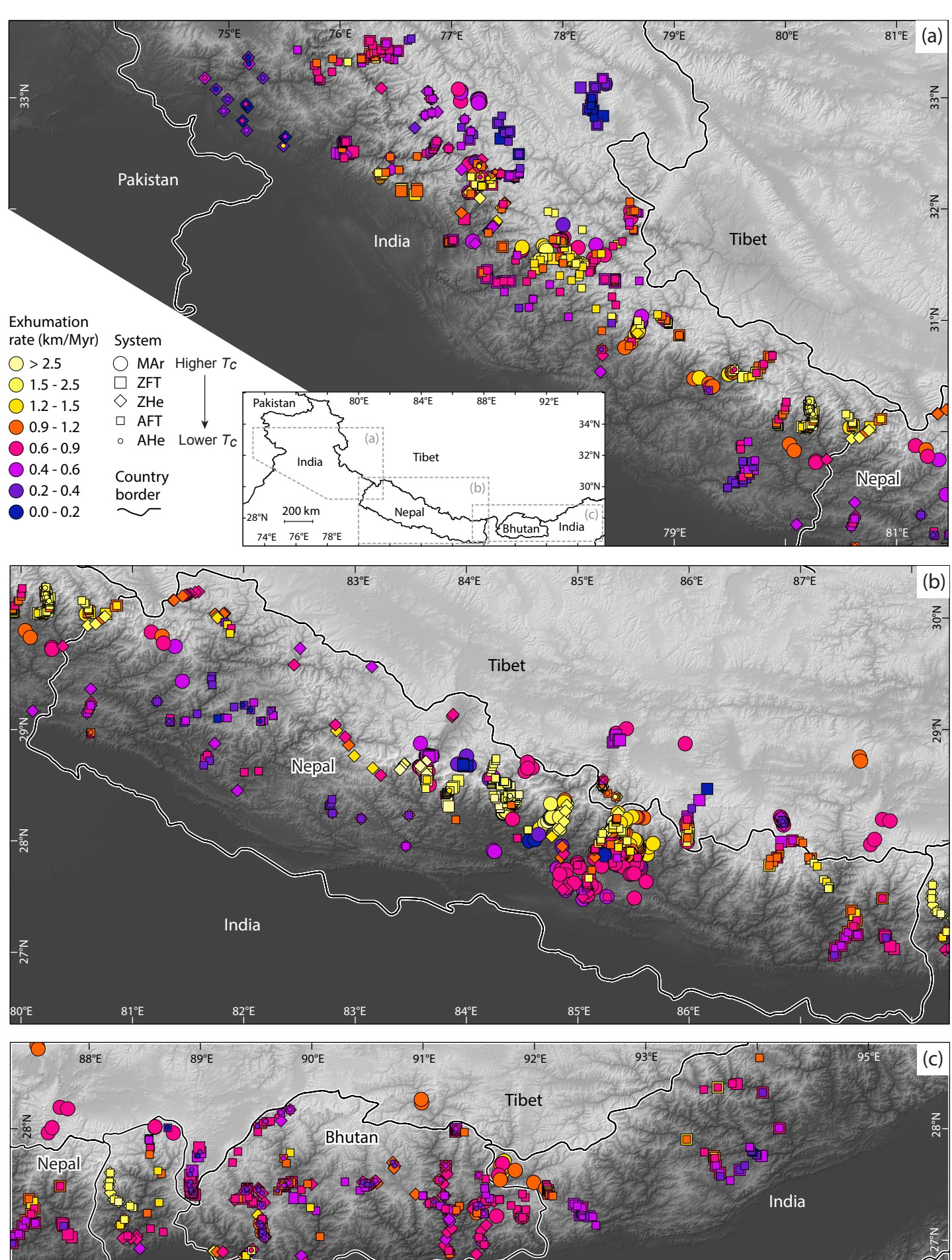

van der Beek and Schildgen
Figure 4

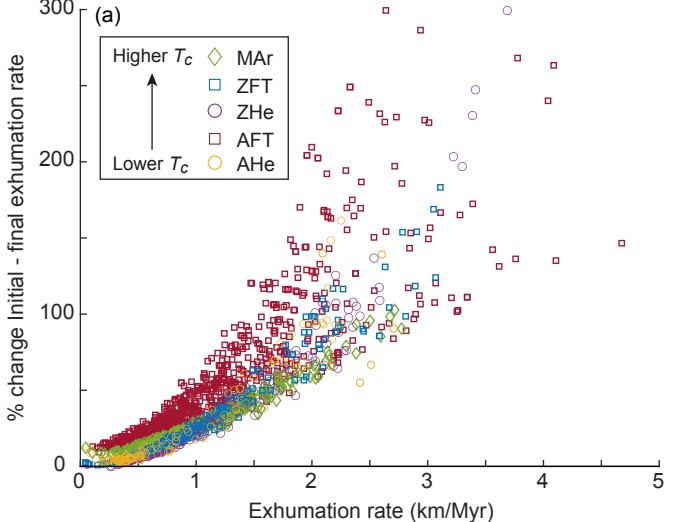

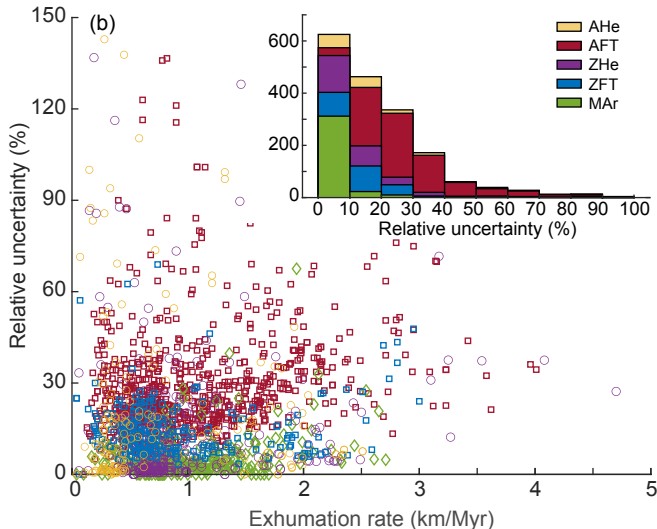

van der Beek and Schildgen
Figure 5

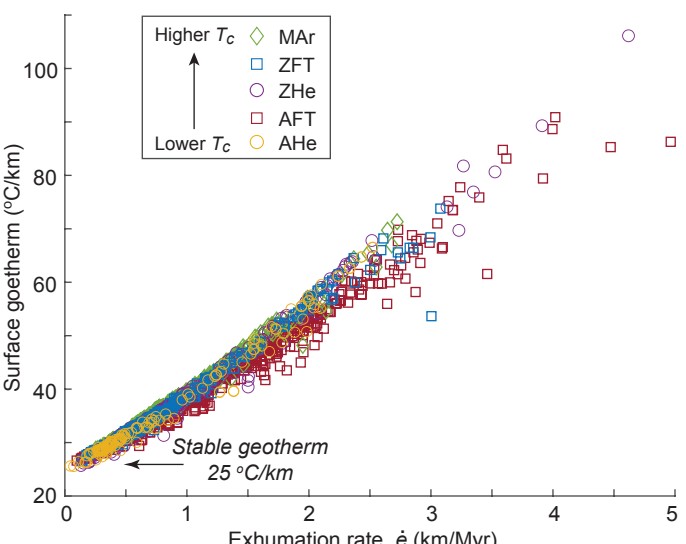

van der Beek and Schildgen
Figure 6

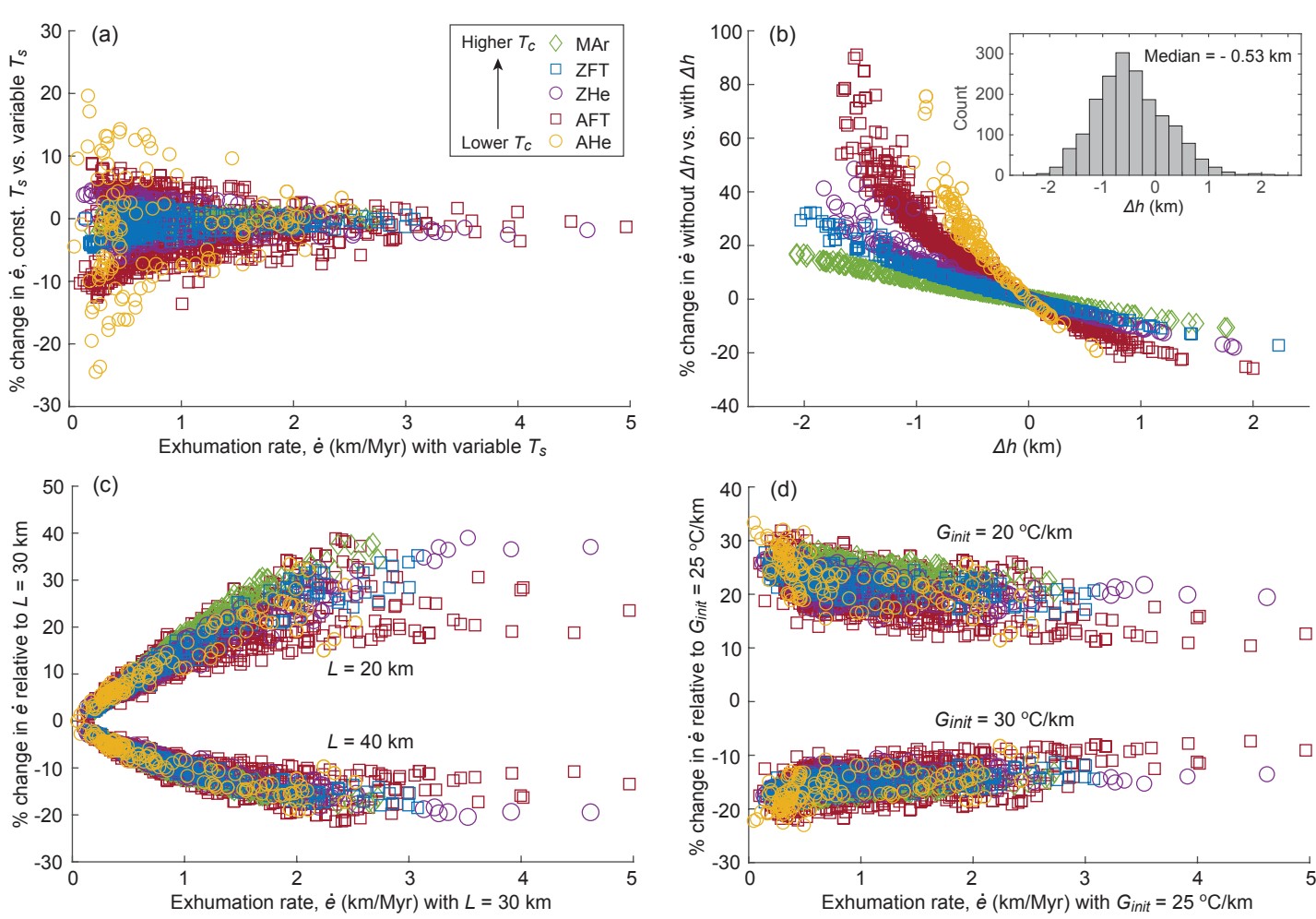

van der Beek and Schildgen
Figure 7