# Peer review of "Short Communication: age2exhume - A Matlab/Python script to calculate steady-state vertical exhumation rates from thermochronometric ages and application to the Himalaya"

_EGUsphere, 2022_

## Author Comment (AC1)

We thank reviewer Matthew Fox for his constructive and helpful comments on our manuscript. Below, we provide some responses and outline how we incorporate the comments (reviewer comments in blue, our response in black; changes made to the manuscript in *italics*):

The submitted manuscript by Peter van der Beek and Taylor Schildgen provides more detail on code that was published in a paper in 2018 (Schildgen et al., Nature, 2018). The method is already published and does not have any major flaws. Furthermore, it is based on earlier work by Mark Brandon (1997). Therefore, the paper could be published as is.

Technically, the above assessment is not quite correct; although some of the original lines of code were included in the script published by Schildgen et al. (2018), the code we present here is improved substantially in that (1) it calculates denudation rates from ages, rather than vice versa, and (2) it includes a local relief correction and surface temperatures at sample sites. *We have modified the text to make this point clearer.*

However, I would like to highlight some issues with this approach in general and think that mentioning these ideas would strengthen the paper.

The authors use the unperturbed initial geothermal gradient as input for calculating geothermal gradients and the modern geothermal gradient is not really discussed.

Furthermore, there is no way to link samples in space. One of the advantages of codes designed to exploit age-elevation relation relationships (Pecube, QTQt, GLIDE) is that the results should be less sensitive to geothermal gradient and there is a means to estimate the geothermal gradient. Basing the interpretation on an unperturbed geotherm does not provide that advantage and I feel that we are losing useful information.

This is correct, and it is one of the reasons why we emphasize that our code is designed for a synoptic overview of regional variations in denudation rates, and does not replace the codes that do take advantage of information from multiple samples or multiple thermochronometers. We have tried to highlight this point repeatedly in the text (e.g., lines 21-23, 131-133, 323-324 of the original manuscript), as we do not want readers to overlook it.

It is also unclear how well known this unperturbed geotherm can ever be. Or even whether this makes sense. For example, if a sample has a known ZFT age, a geotherm that is consistent with an initial value and exhumation rate can be determined using the code. If this same sample also has a AHe age, why would it make sense to use the initial geothermal gradient before any exhumation as opposed to the geothermal gradient inferred the older ZFT age? This geothermal gradient is surely an improvement over the unperturbed gradient.

The initial geotherm, importantly, is not a geotherm that characterizes the model some time prior to the closure of any given thermochronometer, but rather it is the **unperturbed** geotherm. *We have modified the text to make this point clear to readers*. It has an impact on the final solution because it sets the thermal state of the model **prior to** any exhumation (for example, approximating the effect of heat production within the crust, as this is not explicitly included in the model). This means that we cannot input

later, adjusted (or partially adjusted) geotherms without changing the final geotherm, and thus, the final inferred exhumation rate. It could be considered a limitation of the approach we use, but it enables the rapidly converging analytical solution to a steady-state geotherm that the model provides. Theoretically, the unperturbed geotherm could be reasonably well estimated if sufficient information on the regional crustal and lithospheric structure is available.

This concept raises the issue that a single sample may predict different exhumation rates for the modern period but also completely different geothermal gradients. This means that if we are lucky enough to have an estimate of the modern geothermal gradient close by, we would have lots of internally inconsistent pieces of evidence on the exhumation rate. We would also have no way forward with this steady state approach.

It is correct that the steady-state geotherm will change if there are multiple thermochronometers for a single sample that yield different average exhumation rates. However, this steady-state geotherm is not equivalent to the modern geotherm – they would only be equal if exhumation has indeed been steady since the closure of that particular thermochronometer. However, there are very few estimates of the modern geotherm available for mountain belts. For instance, the global heat-flow database (https://ihfc-iugg.org) contains no data whatsoever for the entire Himalaya; although there are data both for the Tibetan Plateau to the north and the Ganges foreland basin to the south, these are not useful for assessing the perturbed geotherm within the mountain belt. Therefore, at least in this part of the world, the issue is irrelevant.

As we have emphasized, in areas of changing exhumation rates, our code should be used to highlight where more advanced thermal models should be used to obtain accurate results with non-steady state solutions (see lines 216-219, 238-239, 244-247 of the original manuscript). Areas of non-steady exhumation violate the underlying assumption of the code, which is why it should only be used to provide synoptic overviews of exhumation rates.

The paper is very critical of GLIDE (Fox et al., Esurf, 2014) and does not really highlight our response to the earlier criticism of Schildgen et al., (Nature, 2018). For example, on line 125 please consider changing the language from "Moreover, it has been shown that the code translates abrupt spatial variations in thermochronological ages, such as across faults, into temporal increases in exhumation rates (Schildgen et al., 2018)." To "Moreover, it has been argued that the code translates abrupt spatial variations in thermochronological ages, such as across faults, into temporal increases in exhumation rates (Schildgen et al., 2018)." This is because, we have argued that this is not the case in Willett et al., (Esurf, 2020). Also, Fox and Carter (Geosciences, 2020) argued that Schildgen et al. (2018) did not show that most sampled regions on Earth may have insufficient data coverage for unbiased prediction of exhumation-rate histories using GLIDE. Instead, the synthetic data produced for analyses carried out by Schildgen et al., (2018) significantly changed the temporal and spatial resolution of the data with respect to the real data. In other words, the spatial and temporal resolution for the real ages from the Western Alps is sufficient because there are old and young ages either side of the fault. In contrast, the spatial and temporal resolution for synthetic ages from the Western Alps is insufficient because of the dramatically different age distributions. This is a complicated issue and it is not appropriate to simplify it to such a degree here.

*We changed the sentence, as suggested, as well as our remaining description of GLIDE to better explain where GLIDE generally works well and where it runs into limitations.*

We feel that the point of the synthetic tests shown by Schildgen et al. (2018) was mischaracterized in Willett et al. (2020) and Fox and Carter (2020). The tests show that for a range of simple tectonic scenarios and typical data distributions, data will commonly be insufficiently resolved to avoid producing spurious increases in exhumation rates over time. The point was not to attempt a detailed modeling of the specific sites used as examples of typical data distributions. Nevertheless, any argument that data are in fact well enough resolved in the western European Alps (the most data-dense location in the world) to avoid this bias has no bearing on whether data in other parts of the world are well enough resolved to avoid the bias. Unfortunately, resolution cut-off values used for many published applications of GLIDE have been far too low to justify ignoring the issue, and guidance on what resolution values are sufficient in cases where data density may be sufficient remains vague. In any case, the point has been discussed rather extensively elsewhere (see Discussion associated with Willett et al., 2021, ESurf), and is beyond the scope of the current manuscript.

Finally, on line 129, the authors argue that GLIDE is as slow as Pecube. However this is almost certainly not the case if the areas are the same size and if Pecube is run in inverse mode. In fact, this is one of the reasons we developed GLIDE. GLIDE does not use the same half space solution as the Willett and Brandon code. It uses a numerical model with a flux boundary condition. Changing this to a fixed boundary condition requires uncommenting one or two lines.

Our own experience with GLIDE is that set-up time and model run time were broadly equivalent to similar (forward) models run in Pecube. However, we agree that running inverse models in Pecube is much slower *and have therefore removed this argument from the text. We also corrected the description of the thermal model used by GLIDE.*

It is unclear how \delta h is actually calculated. Where does \delta h appear in the flow chart in Figure 1B? This is crucial because the closure depth can change by over 100% during the iterative process. How is this accounted for during the whole process?

The initial estimate of $\Delta h$ is based on a nominal initial estimate of closure depth, which is not iteratively adapted (an approach similar to Willett and Brandon, 2013). *We have clarified this point in the text.* Overall, we have found that substantial changes in the initial closure depth (by several hundred meters to 1 km for deeper closure depths) used to estimate $\Delta h$ have little effect on the final result, so we believe this approach is reasonable, as it preserves the ability to re-run the code rapidly without the need for an input digital elevation model and associated operations. *We modified Figure 1B to illustrate that $\Delta h$ is introduced as an input parameter to the modeling.*

The code requires that users extract \delta h from a different piece of software before the code can be used. It is unclear how long this takes so it is unfair to argue that the approach presented here is particularly fast when it is unclear how long it takes to run the other software.

*We have clarified in the text that $\Delta h$ is calculated for the full data set independently of the thermal model, as $\Delta h$ only needs to be determined once* (and included in the input

file), whereas a user may opt to run the thermal model with many different combinations of thermal/site parameters, particularly if they want to explore the sensitivity of the results to those parameters. We assume users will calculate $\Delta h$ with whatever tools they find most convenient, e.g., any GIS software, GMT, etc. The time to do so will depend on the user; using standard GIS functions (as outlined in the supplement) will very likely require under an hour for a basic user (e.g., anyone with basic training in GIS-based DEM analysis) to apply, or tens of minutes for a more advanced user. Once this information is obtained for samples in the input dataset, it need not be re-extracted.

I don't understand why the upper boundary condition of the thermal model has a temperature evaluated at the sample elevation. Surely the thermal model should have an upper temperature boundary condition equated at h\_{ave}. Much more information is required to actually understand this part of the code. For example, if you have two samples from exactly the same geographic position but separated vertically by 2km, what is the thickness of the thermal model? If the surface temperature is used for the two different thermal models, the geothermal gradients will be different for the whole thickness of the crust. Why would the closure elevation (with respect to the centre of the earth) depend on the sample elevation? Given that most data are collected with the aim of producing age-elevation relationships, this needs to be discussed and considered.

This is a very good suggestion; we agree with the reviewer that this is a better approach than the one we initially included, particularly as it results in deeper isotherms being less affected by surface-temperature variations compared to near-surface isotherms, as we would expect. *We have modified the code, the results, and all derived sensitivity tests (plus their descriptions in the main text) accordingly.*

*We have also modified the way that $T_L$ is calculated, using the upper temperature boundary set to a temperature that is equivalent to the average elevation of the entire dataset.* This prevents the scenario that the reviewer describes, with different geotherms predicted for samples separated vertically from one another.

While these changes are important, as they make for a more theoretically sound approach, the impact on calculated exhumation rates is relatively small; we find that a vast majority of the rates changed by less than 5% following these modifications to our code.

In addition, it is not clear why a constant thermal model thickness is used for the whole of the Himalayas. Surely if L represents a crustal thickness, the model should get much thicker as the topographic elevations increase to account for isostasy.

Yes, this is a clear oversimplification, as is the assumption of vertical exhumation pathways, both of which enable rapid model calculations. The former could be addressed by running different parts of the dataset with different *L* values; our sensitivity test showing how much exhumation rates change based on different *L* values (Fig. 6d) helps to illustrate the potential magnitude of this effect for different thermochronometers over a wide range of exhumation rates. We provide the full dataset to users in the Zenodo repository (already properly formatted as an input file) so that they are free to test whatever parameter values they choose.

One other point we want to better emphasize in the text is that *L* is not the crustal thickness, but rather the model thickness. In the case of the Himalaya, that could be considered equivalent to the thickness of the overriding plate, which certainly does change both along and across strike. *We have modified the text to clarify this point; this description now appears at the start of the section describing the age2exhume method.*

There is a lot of discussion on thermal boundary conditions. I don't think detrital thermochronology tells us anything about how appropriate a fixed temperature at 30 km depth is over 10s of millions of years. Since the earliest studies in the Alps by Wagner, it has been shown that the focus of exhumation has been variable. Vernon et al., (EPSL, 2008) also showed that it was variable using iso-age surfaces. Even the results presented in Figure 4 show that exhumation rates have changed at specific locations. A constant lag time is more likely telling us that there is an area that is eroding at a common rate through time, but that that area might be variable. This seems reasonable given that the maximum erosion rates likely depend on long term tectonics and regional climate.

The inherent problem with a half-space model is that it continues to advect heat upwards over time, which is not reasonable over long timescales. Regardless of what exactly a constant lag time shows us (which is a different issue), a half-space model will not correctly reproduce a constant lag time for a detrital dataset without a continuously decreasing exhumation rate that perfectly offsets the advection of isotherms. The reviewer focuses his comments on detrital thermochronology data to those obtained in the European Alps; however, there are also datasets from spatially more restricted regions (e.g., Namche Barwa syntaxis, coastal range of Taiwan, west flank of the southern Alps of New Zealand) that show the same behavior and where the argument of temporally varying loci of maximum exhumation is much more difficult to make.

The thermal half space solution was also used by Moore and England 2001. They elegantly showed that by accounting for transient advection of heat, data that had been interpreted as recording accelerating erosion rates, were actually consistent with a constant exhumation rate. This study is not discussed at all but this result highlights the need to account for transient advection of heat. Without accounting for this, there will be the possibility of a systematic bias towards inferring accelerating exhumation rates as geotherms become higher during exhumation.

We completely agree with the above statement but do not see how it is relevant to our study. It appears that the reviewer may be confusing **transience** with **advective perturbation** of the geotherm. Moore and England (2001) showed that data inferred from **different thermochronometers in the same sample** will often show more rapid cooling for the lower-temperature system and that this can be explained by advective perturbation of the geotherm. This is exactly the effect that is taken into account here. Moore and England (2001) focus on the transient solution and use the well-known expression derived by Carslaw and Jaeger (1959) for the transient advective perturbation of an initially linear geotherm. However, after this initial transient state a thermal steady state **with a perturbed geotherm** is obtained (e.g., Braun et al., 2006) and it is this steady state solution that is employed in our model.

This does not only happen when using a half space solution or a flux boundary condition, it is also predicted with a fixed boundary condition. We also discussed this in Fox and Carter (Geosciences, 2020).

Again, we agree with the above statement: upon the onset of rapid exhumation, thermal models with a fixed-temperature boundary condition will also go through a transient phase where the geotherm is adjusting to the new conditions. What changes between the two thermal boundary conditions is the timescale of transience: from a few million years (depending on the exact thermal parameters) in the case of a fixed-temperature boundary condition to essentially infinite for a thermal half-space model. We checked Fox and Carter (2020) but could not find an instance where this issue was explicitly discussed.

Figure 4. I find the presentation of the results very hard to read. It is almost impossible to visualize whether rates are increasing or decreasing. It also requires people to be familiar with the closure temperatures of the different systems. The fact that a single sample can be associated with 5 different modern exhumation rates is clearly a problem and really defeats the purpose of collecting ages using different systems. Please make a map of the final geothermal gradients as well. How variable are they at a single location and is this a useful result?

Presenting data with high granularity at an orogen scale is challenging within a static figure; nevertheless, we believe there is an advantage of showing the data with its full detail, without any imposed interpolation (how users treat it afterwards is up to them); if users choose to plot results themselves, they can zoom further into specific sites to better examine patterns, particularly in areas where symbols overlap when zoomed out.

*We modified the figure legend to show the order of closure temperatures for the different systems. We also added a sentence to the main text to explain how darker/lighter colors from a lower-temperature system plotted over a higher-temperature system imply decreasing/increasing exhumation rates through time.*

The figure is not meant to illustrate modern exhumation rates per se, but rather steady-state exhumation rates associated with a specific thermochronometric age, with the system represented by the symbol size and shape. When different systems yield different exhumation rates, users should interpret that as indicating a change in exhumation rate through time (see lines 216-219, 238-239, 244-247 of the original manuscript). *We have clarified this point, explaining that exhumation rates are time-averaged rate associated with each thermochronometric age.* The symbols are designed such that if one sample has ages from 5 thermochronometers, the symbols are all stacked, such that the samples with shorter integration times lie above those with longer times, but all are visible (with the caveat that much is hidden areas of very high data density). But as we emphasize (see lines 216-219 and 238-239 of the original manuscript), changes in exhumation rates calculated by the code are always underestimated, and should only be used as a general guide to where changes have occurred – more accurate (non-steady state) exhumation rates can only be obtained with a more advanced thermal model.

At the scale of the whole orogen, which is the scale at which synoptic data are most appropriate, one can use this figure to obtain a *general* picture of the spatial (and to

some extent, temporal) patterns of exhumation rates. Our discussion of the results highlights how general patterns of segmentation across the Himalaya, as well as regions with well-demonstrated changes in exhumation rate through time (e.g., Bhutan), are quite visible within the figure, which we hope readers find to be the case. We believe that for many users, such a synoptic overview can be quite helpful when the aim is to understand general patterns of exhumation, or when one would like to determine more easily where it is worthwhile to spend time on more advanced thermal modeling, or where it might be useful to collect new data.

*We decided against making another map that shows geotherms, as they track nearly linearly with exhumation rates, as we now show in a new figure (reproduced below).*

[Figure]

Line 51: Please provide more details about what the original age2edot actually did. It is unclear whether it actually solved for ages given exhumation rates.

It solves for age given an exhumation rate, as we state on lines 69-73 of the original manuscript (yes, the name of the original version is confusing).

Figures 5 and 6. Please also show the actual values of the variables that are changing. It would be nice to know that a value changes from X to Y as opposed to just the percentage change.

These figures show different information;

In Figure 6a and 6b, we simply show the impact of including or not including a surface-temperature correction and a local relief correction, so there are not specific value-changes that can easily be shown on the figure (it's different for every point).

In Figure 6c and 6d, the values of the changes in variables are shown: in Fig. 6c, a "reference" run with L = 30 km is changed to 20 km and 40 km; in Fig. 6d, a "reference" run with $G_{init}$ = 25 C/km is changed to 20 C/km and 30 C/km. *We have expanded the text of the caption to clarify this point.*

If the reviewer wonders why we show percent changes rather than absolute changes, this is simply because the importance of any change scales with uncertainty, and most uncertainties are in the range of x to x% of the exhumation rate. To illustrate, the figure

below shows a version of Fig. 5 showing absolute exhumation rates, which we find more difficult to interpret than the version included in the manuscript.

---

## Author Comment (AC2)

We thank reviewer David Whipp for his constructive and helpful comments on our manuscript. Below, we provide some responses and outline how we incorporate the comments (reviewer comments in blue, our response in black; changes made to the manuscript in *italics*):

**Summary**

van der Beek and Schildgen presents an overview of a numerical model written in Matlab for the purpose of rapidly calculating estimates of rates of exhumation from regional thermochronology datasets. Their new software uses a steady-state analytical solution to the heat transfer equation with fixed temperature boundary conditions in combination with analytical solutions for estimating thermochronometer ages, which allows for efficient calculation of effective closure temperatures to find exhumation rates that produce the input age(s). The model aims to fill a void in the available options for interpreting thermochronometer age data, providing estimates of exhumation rates for large thermochronometer age datasets (in contrast to models designed for calculating only thermal histories), but with less complexity and computational overhead than other popular software. They demonstrate both the models use for exploring exhumation rates in large datasets and its sensitivity to changes in the input parameters using a large age dataset from the Himalaya, showing major observations from more detailed, local studies are also captured in their model. Overall, the manuscript is clear and well written, and it will certainly appeal to readers of Geochronology. However, there are a few places where the text may be able to be improved in revision, which are detailed below.

**Main comments**

1. There are several places in the text where a choice in the design of age2exhume is suggested to be better than equivalent choices in other models, such as for the boundary conditions. While the authors do justify their reasoning for making such suggestions, model design choices are often based on what is most valid for the physical system being studied. Choosing to use a half-space thermal model with a fixed gradient surface boundary condition will ensure that irrespective of the input exhumation rate, the temperature gradient will remain fixed and honor, for example, borehole temperature measurements / surface heat flow observations. While this is a challenge when working with regional thermochron datasets, it is also an observation that can be used to define the range of allowable solutions that fit the observed age dataset. Because thermochronometer ages are non-unique, it can be otherwise difficult to exclude models that produce the correct ages, but violate other observations. Thus, it would be good to include some text in the discussion that notes some of the limitations of the choice of design in age2exhume for completeness.

We agree that model design choices are (and should be!) based on what is most valid in the particular physical system being studied. A thermal half-space model makes sense when studying wholesale lithospheric thinning, for instance (although note that currently most authors would probably prefer thermal boundary-layer models, see for instance the recent review by Richards et al., PEPI 2020). However, one should distinguish between using a thermal half-space model versus a model with a fixed temperature at the base, and an input stable geothermal gradient versus an input present-day surface geotherm/heat-flow value. These two choices are independent of each other. We agree that, where such information is available, using the present-day heat flow or surface

geothermal gradient will provide additional constraints on possible model outcomes, but reiterate that such constraints are exceedingly rare in mountain belt environments (zero heat flow measurements available in the entire Himalaya for instance). *We have modified the text to make these issues a bit clearer*.

2. Another consideration is the use of Dodson's method to estimate effective closure temperatures. While it is simple and efficient for such calculations, there are also behaviors that are observed in some thermochronometer systems that Dodson's approach does not capture, possibly resulting in difficulties interpreting some age data. For example, it is expected that the effective closure temperature for zircon (U-Th)/He ages will decrease with increasing cooling rate for low-eU zircons (e.g., Guenthner et al., 2013; Whipp, Kellett et al., 2022). This limitation is not discussed, and similar to the comment above, would be a good thing to mention in the discussion section somewhere.

Again, we agree with the reviewer on this point; Dodson's method is a simplification that is strictly valid only in a context of continuous monotonic cooling where temperature evolves inversely with time. This clearly limits applications of this approach to settings where rocks currently at the surface can be reasonably assumed to have cooled monotonously and continuously. The reviewer is entirely correct that more complex settings will not be predicted correctly by the model, also due to the steady-state thermal assumption. *We have included a short discussion of this limitation in the manuscript*.

3. Based on the two comments above, I would suggest adding a section to the Discussion and Conclusions, perhaps titled "Caveats (or assumptions or limitations) and recommended use cases". Here the authors could combine the points related to the limitations of the models to a single section, and also make suggestions for the best applications of the models by future studies. Some of the text from the current section 4.2 could be moved here (lines 317-330), as well as some additions to the text based on the comments above. Perhaps the authors could also comment on the lack of heat production in the geotherm calculation and how that might affect the estimates of exhumation rates. This is not strictly necessary, but may make it easier for users looking for a model to apply to their data to know what the limitations are and be able to easily evaluate whether age2exhume is suitable for their needs. I suspect age2exhume will be a valuable modelling contribution, and more users may choose to utilize the code if they clearly see it would be a good fit for interpreting their data.

*We appreciate this suggestion, and have followed it in revising our manuscript*.

4. Finally, I appreciate the authors making the software source code available but also feel it is worth noting that the choice to provide the software as a Matlab script is somewhat unfortunate. Matlab is a commercial software product, and if the software was available in another language (such as Python or Julia), then it would accessible to a larger user base at no cost. Similarly, it would be nice to have instructions for calculating delta h using freely available GIS software such as QGIS. I am not requesting the authors rewrite their code, but wanted to emphasize there is a limitation in providing Matlab code.

*We have now written a version of age2exhume (the version that takes an input file as an argument) in Python, and will include that, together with an appropriately formatted .csv*

*input file, in a new Zenodo repository referred to in the revised manuscript. We also added a paragraph to the Appendix explaining how users can calculate Δh values and automatically extract them at the location of all sample points in QGIS.*

**Specific remarks (L = line number)**

L15: I know it may not be easy, but is there some kind of clarification that could be used to distinguish a "limited area" from a regional dataset? Is there a certain spatial scale or dataset size that could be used to emphasize when your model should be considered?

A rule-of-thumb guideline for a maximum area that could reasonably be modeled using Pecube with normal modern computing power would be in the order of 100x100 km. *We have added this quantification to the text, but rather in the more detailed description of Pecube (lines 115-120 of the original manuscript) than in the abstract.*

L16: I suppose "largely underutilized" may be somewhat subjective, but perhaps the point here is that it can be challenging to interpret these larger datasets. People have done it (e.g., Herman et al., 2013; Thiede and Ehlers, 2013), but there are not so many tools that are easily applied for such a task (as you say in the introduction). Would it be better to emphasize the challenge and lack of tools here too?

*We have modified the writing here to focus on the challenge of using such large-scale datasets in thermochronology.*

L85: While I can see the authors' point here, particularly for large age datasets, it is also true that failing to consider the constraint the surface heat flow could provide in terms of allowable models that fit the data is also problematic. Heat flow is an observation that can be used to constrain the model parameter space, and it may be practically easier to exclude this due to the paucity of measurements, but that is not necessarily a better solution.

A limitation of our model is that it only calculates steady-state geotherms since the time of closure of the associated thermochronometer, meaning that any changes to the thermal properties of the crust since closure are not considered. Nevertheless, *we have modified the default output of the main version of the code (the one that takes an input file as an argument) to include the surface geotherm*. With that, a user could at least easily compare the calculated steady-state geotherm with an observed modern geotherm.

L89-90: I have not used the code cited here, but based on the plots in the manuscript I cannot see an obvious reason why more rapid exhumation rates would be problematic for code stability. Could you please clarify what the issue is here and possibly why it arises?

That text was not accurately written; *we have clarified now that the age2edot code does not return a solution if the input modern geotherm is far from what would be expected for a high exhumation rate*.

L101-102: This is again an issue about choice of model design and boundary conditions. While the point raised here about a half-space thermal model perhaps not reproducing

constant lag times for regions with a constant rate of exhumation is valid, it is also true that it may be difficult to define a reference temperature in the crust or lithosphere that is fixed over the timescale of orogenic growth and exhumation. For instance, Moho temperatures can change significantly during orogenesis, which is an effect that requires a user to choose an initial geothermal gradient in order to produce a Moho temperature they feel is applicable to the study region. This alone may result in the need to have different initial geothermal gradients for different thermochronometer systems and effective closure temperatures.

*We agree with the reviewer and have added a couple sentences to our new section on "Limitations and caveats" that explains this issue.*

L106: This is a nit-picky thing, but HeFTy is not exclusively an inverse model, but one that can be run as a forward or inverse model.

Correct; *we have adapted the phrasing here*.

L116: "neighbourhood algorithm (Sambridge, 1999)" rather than "natural-neighbor algorithm".

*We corrected this in the text*.

L145: Perhaps it could be clarified that the closure temperature estimate is a value the user should provide, not a calculated value. The text says this, but it may help to note the user should select this as an input value (right?).

The initial estimate of the closure temperature actually has little to no impact on the final solution, as the model rapidly converges to a closure temperature that is consistent with the input age and thermal properties. So, although it can be a user input, we have included default values that we believe suffice. But *we have modified Figure 1 to show input parameters in the workflow, including an initial estimate of closure temperature*.

L148: Would it be better to say that "an estimate of Ts is calculated..." rather than "Ts is estimated..."?

*We have modified this to "Ts is calculated …".*

L168: Is there any justification for the threshold value of the exhumation rate change here?

We experimented with the code to see how many iterations are required for convergence. We found that it is quite fast (most results converge in 7 iterations or less). This enabled us to maintain rapid calculation times even with a rather small threshold for exhumation-rate change. That small value in turn is helpful in ensuring that results are reproducible.

L174: Somewhere in the text above it may be helpful to clarify precisely how the initial geothermal gradient is used, to avoid any possible confusion about it being treated as an initial condition. As it is to me as a reader familiar with modelling, this is clear to me, but it is possible some less familiar readers could be confused. Thus, it may help to state it

is used only to estimate zc at the start of the simulation and to define the basal boundary condition. Just a suggestion, but one that might help readers better understand how the model works.

We had hoped the text and Figure 1 were sufficiently clear on this point but *we have added a phrase to avoid any potential confusion on this point*.

L195-197: This seems like a very useful point and possible way in which new users would really apply the code. Would it be possible to somehow emphasize this point more in the text?

*We have done this by adding a phrase to the abstract and the concluding remarks*.

L216-217: Again, this is somewhat nit-picky, but the model does not make assumptions, the programmer does. In this case, I would suggest rephrasing to say "...time in the model." rather than "...time made by the model".

Agreed, *we made this change*.

L243-244: Is the reference here citable?

Not yet, the manuscript is now in review. However, a pre-print is now available, and we will check the journal guidelines regarding how to appropriately cite it (if we can).

L283-284: This effect is something that has been shown previously in, for example, Mancktelow and Grasemann (1997) and Stüwe et al. (1994). Perhaps it would be note the consistency with earlier work?

While both Stüwe et al. (1994) and Mancktelow and Grasemann (1997) provided important new insights in how to interpret thermochronology data, neither of these included a direct comparison with real-world data. The point we are trying to make here is that the thermal effect of exhumation will affect thermochronometer ages to a degree that exceeds the uncertainty on these ages (i.e., "significantly").

L303-304: I would guess that some readers might not immediately understand the point here about the Peclet number and how changes in L affect the exhumation rate estimates. Could you add a sentence to two to clarify why this happens?

*We have rewritten this sentence to more explicitly state how L affects the modelled thermal structure and thereby the inferred exhumation rates*.

**Figures**

Figures 2 and 3: Would it be possible to produce versions of these figures with a colored contour fill and black contour lines? It may make them easier to read, as the yellow color on a white background is somewhat hard to see. Also, the contour lines in Figure 2 get quite dense along the left side of each panel. Would it be possible to remove some to make it easier to see the remaining lines? Finally, it may be helpful to have a reference gridline for delta h = 0 on the plots.

We appreciate these suggestions to make these two figures more easily readable, and *are experimenting with implementing them*.

Figure 4: Would it be possible to include an orogen-scale inset map somewhere in this figure showing the extents of the panels?

*We have added an inset map that shows where the zoomed panels plot over the scale of the whole orogen. We have made some other slight modifications to the figure to improve readability.*